# LangSAM: Language-Guided Expert Routing on SAM2 for Dense Scene Understanding

## Abstract

Multi-task dense scene understanding tasks require models to jointly reason over heterogeneous visual cues. While foundation vision models like SAM 2 provide strong general-purpose features, their extension to multi-task settings is limited by task interference and the lack of explicit task-aware routing mechanisms. In this paper, we present **LangSAM**, a novel language-guided mixture-of-experts framework built on top of SAM 2 for dense scene understanding. Our key idea is to leverage natural language task prompts to guide expert activation, thereby enabling more effective task-aware feature representation. Specifically, we encode each task prompt and design a text-guided router that fuses the global visual representation with the task embedding to produce task-aware gating signals. These signals are combined with a token-level MoE gate, yielding a dual-gated mechanism that enables experts to specialize both spatially and semantically. To further enhance representation learning, LangSAM incorporates task-specific language-guided MoE blocks for coarse predictions and a shared language-guided MoE block that refines multi-task features by modeling global dependencies. We evaluate LangSAM on two standard datasets, NYUD-v2 and PASCAL-Context, covering six dense prediction tasks including semantic segmentation, depth estimation, human part segmentation, saliency estimation, surface normal estimation, and boundary detection. Extensive experiments show that LangSAM consistently improves over strong SAM2 baselines and recent multi-task learning methods, highlighting the effectiveness of language-guided expert routing as a new paradigm for multi-task dense prediction. The code will be released.

## 1 Introduction

Multi-task dense prediction aims to simultaneously solve multiple pixel-level vision tasks such as semantic segmentation (SemSeg), depth estimation (Depth), surface normal prediction (Normal), human part segmentation (PartSeg), saliency detection (Sal), and boundary detection (Bound) within a single model (Liu et al., 2019; Vandenhende et al., 2020; 2022; Ye & Xu, 2022; Xu et al., 2023a; Tang et al., 2024a). A unified multi-task model not only reduces memory and computation compared to training separate networks but also ensures consistency across outputs and enables holistic scene understanding in applications including autonomous driving and robotics.

Recent progress in large-scale foundation vision models like the Segment Anything Model (SAM) (Kirillov et al., 2023) and its successor, SAM 2 (Ravi et al., 2024), has revolutionized the field by providing powerful, general-purpose visual representations with unprecedented zero-shot capabilities. The natural next step is to leverage these robust visual backbones for multi-task dense prediction, creating a unified model capable of holistic visual scene understanding(Wang et al., 2025). However, when deployed in multi-task learning (MTL) settings, SAM 2 remains fundamentally task-agnostic: the same strong representations are indiscriminately shared across tasks, which often leads to interference and limits task specialization. This raises a central challenge: *how can we inject task-aware control into such large backbones without retraining them from scratch?*

Meanwhile, vision-language models (VLMs) such as CLIP (Radford et al., 2021) demonstrate that natural language can serve as a semantic prior, encapsulating high-level task intent in an interpretable and flexible form. Recent MTL methods like TaskPrompter (Ye & Xu, 2023b), TaskExpert (Ye & Xu, 2023a), MLoRE (Yang et al., 2024), and SEM (Huang et al., 2024a) attempt to introduce

auxiliary priors or expert routing to reduce interference, but they either rely on handcrafted task-specific modules, parameter-heavy adapters, or low-rank approximations without fully leveraging the semantic richness of language. Thus, the synergy between linguistic task descriptions and expert routing in dense vision remains underexplored.

To address this gap, we propose **LangSAM**, a *language-guided mixture-of-experts (MoE)* framework built on SAM2 for multi-task dense prediction. Our key idea is to use natural language task prompts as routing signals that guide expert selection. Specifically, LangSAM introduces three complementary modules: (1) A **task-specific language-guided MoE**, where a lightweight router conditions expert activation on both visual features and task embeddings derived from frozen CLIP encoders. This design enables fine-grained task-aware feature modulation and reduces task interference. (2) A **shared language-guided MoE**, which captures task-agnostic structures shared across tasks, promoting cross-task transfer and stabilizing training. (3) Residual connections around each MoE block, ensuring that routing enhances rather than disrupts backbone features. Together, these modules allow LangSAM to balance specialization and generalization, yielding interpretable expert behaviors aligned with task semantics.

We evaluate LangSAM on two diverse datasets: NYUD-v2 and Pascal-Context. Results across six dense prediction tasks demonstrate that LangSAM consistently outperforms strong SAM2 baselines and achieves improvements over recent state-of-the-art MTL methods. Importantly, analysis of expert routing distributions reveals that language-guided MoE induces semantically meaningful expert specialization (*e.g.*, depth experts focus on geometry while segmentation experts emphasize semantics), validating our method.

In summary, our contributions are summarized as follows:

- We introduce LangSAM, a novel language-guided mixture-of-experts (MoE) framework for multi-task dense prediction. By leveraging task-specific text prompts, LangSAM integrates language priors into the expert routing process, enabling fine-grained task-aware feature modulation.
- We design a dual MoE architecture consisting of *task-specific language-guided routers* and a *shared MoE*, which jointly balance task specialization and cross-task generalization while preserving stability through residual connections.
- We conduct extensive experiments on NYUD-v2 and PASCAL-Context, where LangSAM consistently improves performance across six dense prediction tasks and surpasses strong state-of-the-art MTL baselines. Moreover, our framework provides interpretable routing behaviors, highlighting the effectiveness of incorporating language signals into multi-task dense prediction.

## 2 RELATED WORK

**Multi-task Learning (MTL) for Scene Understanding.** MTL(Vandenhende et al., 2022) aims to improve the performance and efficiency of models by learning multiple related tasks simultaneously from a shared representation. MTL seeks to exploit the shared structure and correlations among tasks in order to enhance model accuracy, efficiency, and generalization, compared to training separate networks for each task. Despite its potential, applying MTL to dense prediction remains challenging. Key difficulties include mitigating negative task interference and ensuring balanced optimization across tasks with heterogeneous characteristics. To address these issues, recent studies have proposed a variety of architectural innovations and algorithmic strategies. Adapter-based techniques (Bhattacharjee et al., 2023; Liang et al., 2022; Xin et al., 2024; Jiang et al., 2024) insert lightweight trainable modules into frozen pre-trained backbones, enabling parameter-efficient task adaptation. In parallel, task-conditioned methods (Jiang et al., 2024; Huang et al., 2024b; Xu et al., 2023b) and prompt-driven approaches (Ye & Xu, 2023b; Lu et al., 2024) tailor network behavior via task identifiers or learned prompts. For instance, Prompt Guided Transformer (PGT) (Lu et al., 2024) integrates task-specific prompts into self-attention, demonstrating the potential of language cues in task-aware modulation. Diffusion-based MTL approaches such as TaskDiffusion (Yang et al., 2025) and DiffusionMTL (Ye & Xu, 2024) extend denoising processes to jointly reconstruct multiple task labels, effectively capturing cross-task dependencies.

A complementary line of work introduces MoE frameworks for MTL (Shen et al., 2024; Chen et al., 2023a), where routing mechanisms dynamically activate specialized experts to mitigate task interference. Recently, language-guided MoE designs have emerged, leveraging semantic prompts to

steer expert selection and improve interpretability (Zhao et al., 2024). While these methods have advanced the field, they often add significant architectural and still operate on a fixed set of pre-defined tasks. Our method sidesteps the issue of direct gradient conflict by isolating task-specific knowledge into modular experts, thereby providing a scalable and flexible solution to task interference.

**Mixture-of-Experts.** MoE models have emerged as a powerful strategy to balance model capacity with computational efficiency by selectively activating only a subset of experts during training or inference (Jacobs et al., 1991; Jacobs & Jordan, 1993; Tang et al., 2024a;b; Mu & Lin, 2025). An MoE layer consists of two key components: a set of "expert" subnetworks and a "router" or "gating" network that, for a given input, sparsely selects a small subset of experts to activate. This allows for the creation of extremely large models where only a fraction of the parameters are used for any single forward pass, leading to significant efficiency gains. The central idea is to divide the network into multiple expert modules, each capturing a specific subspace of the feature distribution, while a lightweight router dynamically determines which experts should be engaged for a given input.

Although MoE was initially popularized in large-scale natural language processing (Fedus et al., 2022; Du et al., 2022; Dai et al., 2024), recent works have begun to extend this paradigm to multi-task dense prediction (Chen et al., 2023b; liang et al., 2022; Ye & Xu, 2023a; Yang et al., 2024; Jiang et al., 2024). For example, TaskExpert (Ye & Xu, 2023a) represents one of the earliest attempts to integrate MoE within the decoder of an MTL framework, demonstrating its potential to mitigate task interference. Building on this idea, MLoRE (Yang et al., 2024) introduces low-rank experts inspired by LoRA (Hu et al., 2022), thereby reducing parameter overhead while scaling effectively to multiple tasks. Beyond pure MoE designs, task-conditioned adapters (Jiang et al., 2024; Han et al., 2024) provide an alternative that inserts parallel adapter modules conditioned on task prompts. Although not explicitly framed as MoE, such designs echo the gating mechanism of expert selection, highlighting the convergence between adapter-based and MoE-based approaches in MTL (Ye & Xu, 2023a). To the best of our knowledge, this work is the first to employ natural language task prompts as explicit routing signals in a unified dense prediction model. While prior language-conditioned approaches typically use text to modulate visual features or to guide generative processes, our method directly leverages language to control an MoE router for discriminative MTL. This design effectively bridges instruction-following with efficient sparse architectures, marking a novel paradigm for task-aware computation.

## 3 METHODOLOGY

### 3.1 BACKBONE REPRESENTATIONS AND TASK EMBEDDINGS

**SAM.** The Segment Anything Model is a foundational model for promptable image segmentation, consisting of three primary components: an image encoder, a prompt encoder, and a mask decoder. The image encoder, built upon a Vision Transformer (ViT), extracts high-quality embeddings from input images. The prompt encoder is highly versatile, capable of interpreting diverse user inputs such as points, bounding boxes, and masks. Finally, a lightweight mask decoder efficiently integrates image and prompt embeddings to produce accurate segmentation masks. SAM2 extends this framework by incorporating video inputs and the memory mechanism, while also achieving higher accuracy and faster inference on images. Although the original SAM work briefly discusses the potential of text prompts, the released model does not provide official support for text-based guidance.

**Text Prompt Understanding Tasks.** To explicitly integrate task semantics into expert routing, we transform natural language task descriptions into dense vector embeddings using a pretrained language encoder. For a given task $t$ with description $d_t$ (*e.g.*, assign each pixel a semantic category label from a predefined set of object classes), the procedure is as follows: Tokenization: The description is first tokenized by a pretrained CLIP tokenizer, producing a sequence of tokens suitable for text encoding. Text encoding: These tokens are processed by a frozen CLIP text encoder, which generates contextualized hidden states that capture both syntactic and semantic information. Embedding extraction: We use the global sentence representation from the text encoder as the task embedding, summarizing the semantics of the full description $\ell_t$. This vector is then treated as the task embedding $e_t \in \mathbb{R}^{512}$. Task embedding dictionary: All embeddings are stored in a dictionary $t : e_t$, providing fixed semantic representations for each task. These embeddings serve as language-guided control signals for conditioning the routing mechanism in task-specific language-guided MoE.

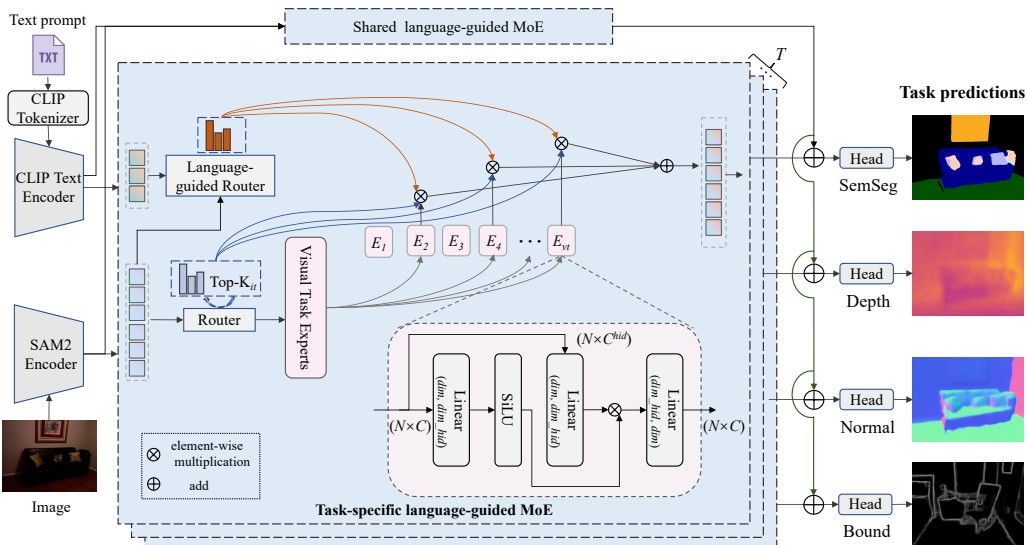

Figure 1: Overview of our LangSAM model. Given an input image, the SAM2 encoder extracts visual representations, while the CLIP tokenizer and text encoder generate task-specific text embeddings. These visual and text features are fused through two complementary modules: (1) a task-specific language-guided MoE, enabling fine-grained expert specialization, and (2) a shared language-guided MoE, which integrates token-level sparse routing with a task-aware language-guided router. The router leverages global task features to compute expert activation weights, thereby endowing the MoE with explicit task-awareness and adaptive routing.

## 3.2 LANGSAM FRAMEWORK

The overview of LangSAM is depicted in Figure 1. The framework is composed of four key components: a CLIP-based text encoder, the SAM2 encoder, task-specific language-guided MoE blocks, and a shared language-guided MoE block. First, the CLIP text encoder converts natural language task descriptions into dense semantic embeddings, providing explicit task-level guidance. Second, the SAM 2 encoder processes the input image to produce generic multi-scale visual features that serve as a shared representation across tasks. Third, the task-specific language-guided MoE blocks fuse SAM2 features with the corresponding task embeddings, generating specialized task features and coarse predictions that are directly supervised by ground-truth annotations. Finally, the shared language-guided MoE block further refines these task-specific representations by modeling cross-task interactions and global spatial dependencies. The refined features are then passed through lightweight task-specific heads to produce the final dense predictions.

## 3.3 TASK-SPECIFIC LANGUAGE-GUIDED MOE

**Revisiting Mixture-of-Experts (MoE).** The MoE framework has emerged as a powerful architecture for scaling model capacity, with early examples such as GShard (Lepikhin et al., 2021) and V-MoE (Riquelme et al., 2021). An MoE layer typically consists of a set of experts $E = \{E_1, E_2, \ldots, E_N\}$ and one or more routers $G = \{G_1, G_2, \ldots, G_M\}$ that determine expert selection. For an input $x$, the router computes gating scores and assigns $x$ to the most relevant experts. The output of the MoE layer can be written as:

$$y = \sum_{i=1}^{N} G_i(x) \cdot E_i(x), \quad (1)$$

where $E_i(x)$ denotes the response of the $i$-th expert and $G_i(x)$ is the corresponding routing weight.

In practice, the gating function is often implemented as a softmax over a linear projection of $x$,

$$G_i(x) = \text{Softmax}(xW_g), \quad (2)$$

with $W_g$ being a learnable parameter matrix. This formulation allows the router to dynamically select and weight experts according to the input representation, thereby enabling conditional computation and efficient scaling of model capacity.

To enable task-aware specialization on SAM2 features, we design a language-guided MoE block that combines token-level sparse routing with task-level language guidance. The block consists of three key components: a visual router, a language-guided router, and a set of experts. A gate computes token-wise weights and top-$k$ expert indices based on visual features, following recent MTL MoE method (TaskExpert (Ye & Xu, 2023a)). Simultaneously, the language-guided router provides global, task-aware coefficients for each expert. During expert computation, outputs are modulated by both token-level weights and task-level language weights. This synergy allows fine-grained specialization (per-token routing) while maintaining global task consistency (per-task language routing).

Let $F_v$ and $e_t$ denote the task-specific language-guided MoE block inputs, where $F_v$ is the visual feature and $e_t$ is text feature. Formally, the output of the task-specific language-guided MoE is calculated as:

$$y_{ts} = \sum_{i=1}^{N_{vt}} G_i(F_v) E_i(F_v) G_{lang_i}(F_v, e_t), \tag{3}$$

$$G_i = \frac{g_i}{\sum_{j=1}^{N_{vt}} g_j}, \tag{4}$$

$$G_{lang_i} = \frac{g_i^{lang}}{\sum_{j=1}^{N_{vt}} g_j^{lang}}, \tag{5}$$

$$g_i = \begin{cases} s_i, & s_i \in \text{Top-k}(\{s_j | 1 \le j \le N_{vt}\}, K_{vt}) \\ 0, & otherwise, \end{cases} \tag{6}$$

$$s_i = \text{Sigmoid}(F_v^\top e_i), \tag{7}$$

where $N_{vt}$ denotes the number of visual task experts. $g_i$ denotes the gate value for the $i$-th expert. $s_i$ is the visual feature to expert affinity. $e_i$ is the centroid of the $i$-th expert. Top-$k(s_i,K)$ represents the set of $K$ highest affinity scores calculated for visual task feature and routing experts. The visual task expert design increases the number of non-zero gates to $N_{vt}$, enabling full activation of all task-specific experts.

**Language-guided router.** To effectively integrate task semantics into the expert routing process, we introduce a *language-guided router*, as shown in Figure 2. Unlike conventional MoE gating that relies solely on visual features, our router incorporates both vision and task-specific textual embeddings to produce routing weights. Given the input visual features $F \in \mathbb{R}^{H \times W \times C}$, we apply global average pooling across spatial dimensions to obtain a compact visual descriptor:

$$\mathbf{F_p} = \text{MeanPool}(F_v) \in \mathbb{R}^C. \tag{8}$$

In parallel, we encode the text prompt of the task description into an embedding vector $e_t \in \mathbb{R}^d$ using a pretrained text encoder (*i.e.*, CLIP).

Specifically, given a pooled visual feature $\mathbf{F_p} \in \mathbb{R}^C$ from the backbone and a task embedding vector $\mathbf{e_t} \in \mathbb{R}^d$ obtained from the text encoder, we first project them into a joint latent space:

$$\mathbf{h} = \tanh(W_p \mathbf{F_p} + W_t \mathbf{e_t}), \tag{9}$$

where $W_p$ and $W_t$ are learnable projection matrices. The joint representation $\mathbf{h}$ is then mapped to expert logits through a linear layer, followed by a softmax normalization to obtain the routing distribution:

$$g^{lang} = \text{Softmax}(W_o \mathbf{h}). \tag{10}$$

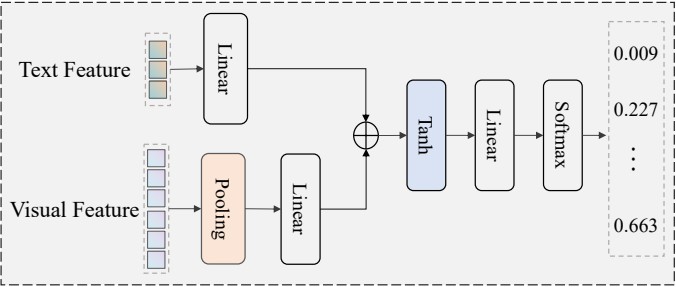

Figure 2: Overview of our language-guided router. Given text and visual features, the router maps them into a joint space and outputs task-aware routing weights, which activate a subset of experts for dynamic feature modulation.

Here, $g^{lang} \in \mathbb{R}^E$ encodes the probability of assigning the input features to each of the $E$ experts. By conditioning the routing on both vision and text cues, the language-guided router explicitly aligns expert selection with task semantics, enabling the model to dynamically activate the most relevant experts for each task while maintaining cross-task consistency. This design provides a principled mechanism to leverage language priors for task-aware routing, which is crucial in MTL where different tasks require specialized feature transformations.

## 3.4 SHARED LANGUAGE-GUIDED MoE

In addition to task-specific language-guided MoE blocks, we introduce a *shared language-guided MoE* that provides a global mechanism for cross-task feature refinement. While task-specific MoEs focus on learning specialized transformations aligned with individual task prompts, the shared MoE leverages language-conditioned routing to capture common semantic structures that benefit all tasks. Formally, given a visual feature $\mathbf{F_v}$ and task embedding $\mathbf{e_t}$, the shared MoE computes a language-aware routing distribution that activates a subset of experts,

$$y_s = \sum_{i=1}^{N_{vt}} G_i(\mathbf{F_v}) E_i(F_v) G_{lang_i}(\mathbf{F_v}, \mathbf{e_t}), \tag{11}$$

where $G_i(\mathbf{F_v})$ and $G_{lang_i}(\mathbf{F_v}, \mathbf{e_t})$ denotes the language-conditioned routing weights for expert $i$. By aggregating information across tasks through a shared pool of experts, this design enables knowledge transfer and regularization, effectively preventing overfitting to a single task while enhancing the consistency of multi-task representations. In practice, the shared MoE acts as a global bridge, complementing task-specific MoEs with cross-task inductive biases and ensuring that the framework exploits both task-level specialization and global task-agnostic knowledge.

We introduce a residual connection by directly adding the original feature $\mathbf{x}$ back to the aggregated output $\mathbf{y}$. Specifically, the update rule is formulated as:

$$\mathbf{y} \leftarrow y_{ts} + y_s. \tag{12}$$

This residual pathway stabilizes training by preserving the original feature information, thereby preventing the degradation of task-aware representations caused by overly aggressive expert routing.

## 3.5 TRAINING OBJECTIVE

Training the proposed LangSAM involves optimizing a multi-objective loss that simultaneously balances task-specific objectives, encourages effective expert specialization, and facilitates beneficial cross-task knowledge transfer. The overall training objective is formulated as the multi-task loss:

$$\mathcal{L}_{mt} = \sum_{t=1}^{T} \beta_t \mathcal{L}_t, \tag{13}$$

where $\beta_t$ denotes the set of hyperparameters (*i.e.*, balancing factors) and $\mathcal{L}_t$ represents the task-specific losses, and $T$ is the total number of tasks with $t \in [1, T]$.

## 4 EXPERIMENTS

**Datasets.** We evaluate LangSAM on two standard benchmarks: NYUD-v2 (Silberman et al., 2012) and PASCAL-Context (Chen et al., 2014). NYUD-v2 consists of 1,449 annotated RGB-D images from 464 indoor scenes, split into 795 training and 654 testing samples, supporting tasks including semantic segmentation (SemSeg), depth estimation (Depth), surface normal prediction (Normal), and boundary detection (Bound). PASCAL-Context provides dense pixel-level annotations for 10,103 images (4,998 train / 5,105 val), covering SemSeg, human part segmentation (PartSeg), saliency detection (Sal), Normal, and Bound tasks.

**Baselines.** We compare the proposed LangSAM with several baselines, including single-task baselines (STB), multi-task baselines (MTB), and hard-parameter sharing (HPS).

**Evaluation metric.** We adopt six evaluation metrics to comprehensively assess the performance of multi-task models. (1) Mean Intersection-over-Union (mIoU) is employed to evaluate both semantic segmentation and human part segmentation. (2) Root Mean Square Error (RMSE) is used to measure the accuracy of depth estimation. (3) Mean Angular Error (mErr) is adopted for surface normal prediction. (4) Optimal Dataset Scale F-measure (ODS-F) is reported for edge detection. (5) Maximum F-measure (maxF) is utilized for saliency estimation. (6) Average per-task performance drop ($\Delta_m$) is introduced as a holistic metric to characterize multi-task trade-offs. Specifically, $\Delta_m = \frac{1}{T} \sum_{i=1}^{T} \frac{F_{m,i} - F_{s,i}}{F_{s,i}} \times 100\%$, where $F_{m,i}$ and $F_{s,i}$ denote the performance of the multi-task model and the single-task baseline on task $i$, respectively, and $T$ is the number of tasks. A higher $\Delta_m$ indicates better overall multi-task performance.

**Implementation details.** The backbone is the pretrained SAM 2 encoder, which is frozen during training to preserve its general-purpose dense representations. The multi-scale features from SAM 2 are upsampled and concatenated to form a feature map of dimension 352, which is used as the input to the task-specific modules. We set the batch size to 2 for NYUD-v2, and 2 for PASCAL-Context. The number of epochs is set to 200 for NYUD-v2, and 30 for PASCAL-Context. We adopt the Adam optimizer with an initial learning rate of $1 \times 10^{-3}$, a linear warmup of 5% of training steps, and weight decay of $1 \times 10^{-6}$. For loss functions, we adopt cross-entropy loss for SemSeg and PartSeg, L1 loss for Depth and Normal, and Balanced Binary Cross Entropy Loss for Bound. **Text Prompts.** For each dense prediction task (SemSeg, Depth, Normal, Bound, PartSeg, Sal), we design short natural language descriptions. These prompts are encoded by the pretrained CLIP text encoder (ViT-B/32) without fine-tuning. The [CLS] token embedding is used as the task embedding and cached for efficient training. **Task Heads.** For each task, we attach a lightweight convolutional decoder with one hidden layer of dimension equal to the final embedding size. Each head upsamples the expert-aggregated features to the original input resolution using bilinear interpolation.

### 4.1 RESULTS

**Results on PASCAL-Context.** Table 1 reports results on the PASCAL-Context benchmark across five dense prediction tasks. When using the ViT-L backbone, LangSAM achieves strong overall performance, reaching 81.91 mIoU on SemSeg, 69.42 mIoU on PartSeg, 84.92 maxF on Sal, 13.51 mErr on Normal, and 74.20 odsF on Bound, while requiring fewer FLOPs and parameters than recent competitors such as TaskPrompter, TaskExpert, and SEM. These results indicate that incorporating language-guided routing improves both efficiency and accuracy, particularly in balancing semantic and geometric tasks. With the SAM2-H-L backbone, LangSAM further demonstrates its adaptability, achieving competitive results with notable gains on PartSeg and Bound, underscoring the framework's scalability to large foundation models. Visual results in Appendix Figure 7.

**Results on NYUD-v2.** As shown in Table 2, most previous methods (*e.g.*, InvPT (Ye & Xu, 2022), TaskPrompter (Ye & Xu, 2023b), TaskExpert (Ye & Xu, 2023a)) fall short of the single-task baseline, indicating severe task interference. MLoRE (Yang et al., 2024) alleviates this issue slightly with low-rank experts, but only achieves marginal gains (+0.11%). In contrast, LangSAM delivers consistent improvements across tasks, notably reducing depth error (rmse 0.4701) and normal error (mErr 16.72), while also achieving the best boundary detection (odsF 79.10). Overall, LangSAM yields a clear average gain of +3.66%, demonstrating the benefit of language-guided routing in mitigating task conflicts and enhancing multi-task dense prediction. Visual results in Appendix Figure 8.

Table 1: Experimental results on PASCAL-Context dataset. '↓': lower is better. '↑': Higher is better. $\Delta_m$ denotes the average per-task performance drop (the higher, the better).

| Model | Backbone | SemSeg (mIoU)↑ | PartSeg (mIoU)↑ | Sal (maxF)↑ | Normal (mErr)↓ | Bound (odsF)↑ | $\Delta_m[\%]$↑ | FLOPs (G)↓ | Params (M)↓ |
|---|---|---|---|---|---|---|---|---|---|
| STB | HRNet18 | 62.23 | 61.66 | 85.08 | 13.69 | 73.06 | 0.00 | - | - |
| MTI-Net | HRNet18 | 61.70 | 60.18 | 84.78 | 14.23 | 70.80 | -2.10 | 161 | 128 |
| ATRC | HRNet18 | 57.89 | 57.33 | 83.77 | 13.99 | 69.74 | -4.45 | 216 | 96 |
| DeMT | HRNet18 | 59.23 | 57.93 | 83.93 | 14.02 | 69.80 | -3.79 | - | - |
| STB | ViT-L | 81.62 | 72.21 | 84.34 | 13.59 | 76.79 | 0.00 | - | - |
| PAD-Net | ViT-L | 78.01 | 67.12 | 79.21 | 14.37 | 72.60 | -5.72 | 773 | 330 |
| MTI-Net | ViT-L | 78.31 | 67.40 | 84.75 | 14.67 | 73.00 | -4.62 | 774 | 851 |
| ATRC | ViT-L | 77.11 | 66.84 | 81.20 | 14.23 | 72.10 | -5.50 | 871 | 340 |
| InvPT | ViT-L | 79.03 | 67.61 | 84.81 | 14.15 | 73.00 | -3.61 | 669 | 423 |
| TaskPrompter | ViT-L | 80.89 | 68.89 | 84.83 | 13.72 | 73.50 | -2.03 | 497 | 401 |
| TaskExpert | ViT-L | 80.64 | 69.42 | 84.87 | 13.56 | 73.30 | -1.74 | 622 | 420 |
| SEM | ViT-L | 81.66 | 69.90 | 84.95 | 13.39 | 73.80 | -0.98 | - | - |
| LangSAM (Ours) | ViT-L | 81.91 | 69.42 | 84.92 | 13.51 | 74.20 | -0.87 | 479 | 301 |
| LangSAM (Ours) | SAM2-H-L | 77.33 | 73.85 | 84.58 | 13.71 | 77.10 | -0.57 | 361 | 252 |

Table 2: Experimental results on NYUD-v2 dataset. '↓(↑)': lower (higher) is better.

| Model | SemSeg (mIoU)↑ | Depth (rmse)↓ | Normal (mErr)↓ | Bound (odsF)↑ | $\Delta_m[\%]$ |
|---|---|---|---|---|---|
| Single task | 56.77 | 0.5141 | 18.56 | 78.93 | 0.00 |
| InvPT | 53.56 | 0.5183 | 19.04 | 78.10 | -2.52 |
| TaskPrompter | 55.30 | 0.5152 | 18.47 | 78.20 | -0.81 |
| TaskExpert | 55.35 | 0.5157 | 18.54 | 78.40 | -0.84 |
| MLoRE | 55.96 | 0.5076 | 18.33 | 78.43 | 0.11 |
| LangSAM (Ours) | 54.36 | **0.4701** | **16.72** | **79.10** | **3.66** |

Table 3: Fine-tuning results on PASCAL-Context. We adopt a straightforward fine-tuning setup where the SAM 2 backbone is frozen, following the evaluation protocol of MTSAM.

| Model | SemSeg (mIoU)↑ | PartSeg (mIoU)↑ | Sal (mIoU)↑ | Normal (mErr)↓ | Params (M) | $\Delta_m$↑ |
|---|---|---|---|---|---|---|
| HPS | 64.77 | 57.91 | 64.10 | 14.21 | 30.07 | +0.00% |
| STL | 65.14 | 58.58 | 65.02 | 15.94 | 63.60 | -2.25% |
| Cross-Stitch | 64.97 | 58.63 | 64.46 | 15.32 | 79.46 | -1.42% |
| MTAN | 64.56 | 59.08 | 64.57 | 14.74 | 36.61 | -0.33% |
| NDDR-CNN | 65.28 | 59.18 | 65.09 | 15.57 | 69.25 | -1.26% |
| HyperFormer | 71.43 | 60.73 | 65.54 | 17.77 | 287.32 | -1.91% |
| Polyhistor | 70.87 | 59.54 | 65.47 | 17.47 | 34.18 | -2.14% |
| Polyhistor-Lite | 70.24 | 59.12 | 64.75 | 17.40 | 11.29 | -2.72% |
| LoRA-HPS (r=32) | 48.19 | 46.73 | 69.50 | 20.38 | 74.33 | -19.97% |
| LoRA-STL (r=16) | 55.25 | 71.33 | 75.72 | 17.05 | 86.33 | +1.65% |
| LoRA-STL (r=32) | 65.07 | 72.05 | 76.41 | 16.82 | 110.33 | +6.42% |
| MultiLoRA | 72.39 | 67.78 | 71.66 | 20.07 | 92.80 | -0.16% |
| MTSAM | 74.13 | 71.04 | 76.28 | 17.10 | 74.71 | +8.95% |
| LangSAM(Ours) | 76.01 | 73.48 | 77.02 | 15.91 | 54.10 | +10.88% |

**Results on fine-tuning.** Table 3 shows that classical multi-task baseline methods (*e.g.*, HPS, MTAN) bring limited or inconsistent gains under frozen SAM 2. Recent transformer- and LoRA-based methods improve efficiency but often trade off performance across tasks. MTSAM achieves strong results by introducing task-aware fine-tuning. Our LangSAM further advances the state of the art, delivering the best overall performance (+10.88% $\Delta_m$) with fewer parameters.

## 4.2 ABLATION STUDIES

**Ablation on components.** Table 4 shows that adding task-specific MoE (TLM) already yields clear gains over the multi-task baseline. Introducing task-specific language-guided MoE (TLM) yields substantial gains across all tasks, demonstrating the benefit of language-guided specialization.

Table 4: Ablation study of model components on NYUD-v2 dataset using the SAM 2 backbone. SLM and TLM denote the shared and task-specific language-guided MoE blocks, respectively. LGR denotes the language-guided router.

| Components | SemSeg (mIoU)↑ | Depth (rmse)↓ | Normal (mErr)↓ | Bound (odsF)↑ |
|---|---|---|---|---|
| MTB | 52.23 | 0.4932 | 17.04 | 78.1 |
| *w/* TLM | 53.65 | 0.4867 | 16.70 | 78.9 |
| *w/* TLM+SLM | 54.36 | 0.4701 | 16.72 | 79.1 |
| *w/o* LGR | 53.05 | 0.4807 | 16.80 | 79.0 |

Table 5: Ablation on the number of experts in the language-guided MoE on NYUD-v2 dataset using the SAM 2 backbone. For the other settings, expert is fixed at 6 We select 2, 4, 6, 8, and 16 experts and activated them proportionally.

| Experts | SemSeg (mIoU)↑ | Depth (rmse)↓ | Normal (mErr)↓ | Bound (odsF)↑ |
|---|---|---|---|---|
| 2 | 53.54 | 0.4889 | 16.92 | 78.9 |
| 6 | 54.36 | 0.4701 | 16.72 | 79.1 |
| 8 | 53.28 | 0.4849 | 16.97 | 79.0 |
| 16 | 54.59 | 0.4783 | 16.88 | 79.2 |

Adding the shared MoE (SLM) further improves overall performance, highlighting the complementary role of cross-task knowledge transfer. Removing the language-guided router (LGR) leads to a noticeable drop, highlighting the importance of language cues in expert selection. Overall, these results confirm that each component contributes to the robustness of LangSAM.

**Effect of the number of experts.** As shown in Table 5, performance improves as the number of experts increases from 2 to 6, indicating that a moderate pool of experts effectively captures task diversity. Using too few experts limits specialization, while too many experts (*e.g.*, 16) introduces redundancy and optimization difficulty. The best trade-off is observed around 6 experts, which balances specialization and stability, confirming the importance of properly scaling expert capacity in language-guided MoE. More experiments on expert number in the appendix (see Section A.4.1).

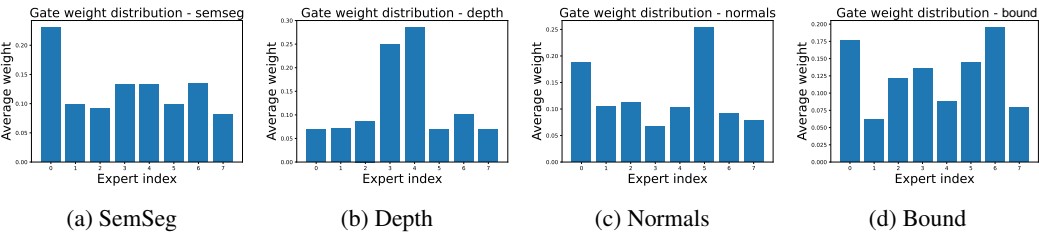

(a) SemSeg     (b) Depth     (c) Normals     (d) Bound

Figure 3: We set the number of experts to 8. Expert routing distributions in **LangSAM** for SemSeg, Depth, Normals, and Boundary detection. Different tasks activate distinct expert subsets, showing that language-guided routing enables interpretable specialization and reduces task interference.

**Effect of the expert specialization.** We visualize the average gating weights across experts for each task (see Figure 3). Interestingly, the distributions vary significantly between tasks: segmentation heavily activates expert 1 and 3, while depth estimation relies more on expert 4 and 5. This indicates that LangSAM learns a meaningful division of labor among experts, where different experts specialize in semantic vs. geometric cues. This validates the core idea of LangSAM: by conditioning expert selection on task descriptions, the model mitigates negative transfer and achieves semantically aligned MTL. We provide a more detailed examination of how experts evolve across different tasks in Appendix A.4.2, offering further insights into task-specific specialization.

## 5 CONCLUSION

We propose LangSAM, a language-guided mixture-of-experts framework that leverages task descriptions as routing signals to inject task-aware control into SAM 2 for multi-task dense prediction. By combining task-specific and shared MoE modules with residual connections, LangSAM achieves a strong balance between specialization and generalization. Experiments on NYUD-v2 and PASCAL-Context show consistent improvements over SAM 2 baselines and recent MTL methods, while expert routing analysis reveals interpretable task-aligned specialization. These results highlight the promise of language as a universal interface for guiding expert computation and point toward future extensions in broader task domains and modalities.

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

## A APPENDIX

Appendix organization:

### A.1 BACKGROUND

#### A.1.1 MULTI-TASK DENSE PREDICTIONS

Dense prediction tasks, such as semantic segmentation, depth estimation, surface normal estimation, and boundary detection, are fundamental to holistic scene understanding. While traditionally studied in isolation, these tasks are highly correlated: geometry (depth, normals) complements structure (semantics, boundaries). Multi-task learning (MTL) therefore offers a unified framework that reduces redundancy and improves generalization by sharing representations across tasks while preserving task-specific specialization.

However, MTL often suffers from negative transfer, where conflicting gradients degrade performance. Existing solutions, including attention mechanisms, cross-task modulation, and adaptive loss weighting, mitigate this issue but remain limited in scalability and interpretability. This motivates more flexible routing strategies, such as mixture-of-experts and language-guided control, which dynamically adapt feature sharing to the needs of each task.

#### A.1.2 MULTI-TASK WITH MOE

Mixture-of-Experts (MoE) has become a promising approach to scale model capacity efficiently by activating only a sparse subset of experts per input. This enables specialization, adaptive capacity allocation, and improved computational efficiency.

For multi-task dense prediction, MoE naturally balances shared and task-specific representations. Instead of forcing all tasks through a single backbone, MoE routes features via distinct experts, mitigating interference and fostering interpretable specialization. Recent works show that such conditional computation outperforms conventional shared-backbone designs by promoting diversity and disentanglement.

However, existing MoE gating is typically data-driven and task-agnostic, leading to routing decisions misaligned with task semantics. Moreover, naively scaling experts increases overhead in multi-task settings. These challenges motivate structured, controllable gating strategies—such as language-guided MoE—that leverage high-level task descriptions for more interpretable and efficient expert selection.

### A.1.3 MULTI-TASK WITH DIFFERENT BACKBONES

The performance of multi-task dense prediction is highly influenced by the choice of backbone architecture. Early approaches relied on CNNs, such as HRNet, which provide strong spatial resolution and remain competitive in dense prediction tasks. With the advent of transformers, Vision Transformers (*i.e.*, ViT-Tiny, ViT-Base, ViT-Large) and hierarchical variants such as Swin Transformer (*i.e.*, Swin-Tiny, Swin-Base, Swin-Large) have demonstrated superior ability to capture long-range dependencies and scale to large datasets, leading to notable improvements in multi-task benchmarks.

More recently, foundation vision models have reshaped the design space for multi-task learning. In particular, the Segment Anything Model (SAM) and its successors offer highly generalizable feature extractors pretrained on large-scale image segmentation tasks. These models provide dense, task-agnostic visual representations that can be adapted to multiple downstream tasks with minimal fine-tuning. However, directly extending such powerful but monolithic backbones to multi-task settings introduces challenges of task interference and insufficient task-specific control.

Therefore, an open research direction is how to effectively integrate different backbone families into multi-task frameworks, leveraging CNNs for spatial detail, transformers for global reasoning, and foundation models such as SAM for strong generalization. Our work follows this direction by adopting SAM 2 as the backbone and introducing a language-guided MoE module to inject task-awareness into its otherwise task-agnostic features.

## A.2 PROOFS

### A.2.1 THEORY: WHY LANGUAGE-GUIDED DUAL GATING IN MoE WORKS

**Setup.** Let $T$ denote a set of dense prediction tasks (*e.g.*, NYUD-v2: SemSeg, Depth, Normal, Bound). Given an image $I$, a backbone (SAM 2) produces a feature $F_v \in \mathbb{R}^{B \times C \times H \times W}$. For task $t \in T$ with language description $\ell_t$, a pretrained text encoder (CLIP) yields an embedding $e_t \in \mathbb{R}^d$. Our MoE block consists of $N$ experts $\{E_i\}_{i=1}^N$ and two gates: (i) a token-level gate $G_{\text{tok}} : \mathbb{R}^C \to \Delta^{N-1}$ that produces top-$k$ sparse weights per location, and (ii) a language-guided global gate $G_{\text{lang}} : \mathbb{R}^C \times \mathbb{R}^d \to \Delta^{N-1}$ that depends on $(\bar{F}_v, e_t)$, where $\bar{F}_v = \frac{1}{HW} \sum_{h,w} F_{V:,h,w}$ is a pooled visual summary. For a token feature $x \in \mathbb{R}^C$, the effective routing weight for expert $i$ is

$$\omega_i(x; e_t) \;=\; G_{\text{tok},i}(x) \,\cdot\, G_{\text{lang},i}(\bar{F}_v, e_t).$$

The MoE output at a token is:

$$y(x; e_t) \;=\; \sum_{i=1}^{N} \omega_i(x; e_t)\, E_i(x).$$

### A.2.2 EXPRESSIVITY AND TASK-CONDITIONAL FACTORIZATION

**Proposition 1** (Task-conditional expressivity). *Assume each expert $E_i$ is a universal approximator on $\mathbb{R}^C$ (e.g., a 2-layer MLP with SiLU function), and both gates $G_{tok}, G_{lang}$ are measurable functions mapping into the probability simplex. Then the induced family $\{x \mapsto \sum_i \omega_i(x; e_t) E_i(x)\}_{t \in \mathcal{T}}$ is a universal approximator for the class of measurable task-conditional functions $\{f_t : \mathbb{R}^C \to \mathbb{R}^C\}$.*

*Sketch.* The classical MoE universal approximation follows by partition-of-unity arguments: gates implement a soft partition of the input space; experts approximate local charts. Here, $G_{\text{lang}}(\cdot, e_t)$ modulates the partition according to $e_t$, thus selecting a *task-conditioned* convex combination of experts. Because $e_t$ varies with $t$, the model instantiates a distinct soft partition per task, and hence can approximate any collection $\{f_t\}$ jointly by reusing experts across tasks when beneficial.

### A.2.3 GENERALIZATION ADVANTAGE VIA SPARSE CONDITIONAL COMPUTATION

**Proposition 2** (Complexity reduction by sparse activation). *Suppose the token-level gate activates at most $k$ experts ($k \ll N$) per token, and experts have parameter norm bounded by $R$. Let $\mathcal{F}_{MoE}$ be the hypothesis class realized by our dual-gated MoE. Then the empirical Rademacher complexity*

*satisfies*

$$\hat{\mathfrak{R}}_n(\mathcal{F}_{MoE}) \sqrt{\frac{k}{N}} \cdot \hat{\mathfrak{R}}_n(\mathcal{F}_{dense}) \quad with \quad \hat{\mathfrak{R}}_n(\mathcal{F}_{dense}) \propto \frac{R}{\sqrt{n}},$$

*where $\mathcal{F}_{dense}$ denotes the class that densely mixes all E experts per token.*

*Sketch.* Because only $k$ experts contribute per token, the effective hypothesis is a $k$-sparse mixture. Standard symmetrization plus contraction yields a complexity scaling with the $\ell_2$-norm of mixture weights. Under a simplex constraint and $k$-sparsity, this norm is upper bounded by $\sqrt{k/N}$ times the dense case, leading to the stated bound. The language gate further *conditions* the mixture on $e_t$, reducing the entropy of the selection and tightening the bound in practice.

**Implication.** Sparse conditional computation lowers the function class capacity relative to dense sharing, which yields *better generalization* at fixed sample size and curbs overfitting due to unnecessary expert co-activation.

A.2.4   NEGATIVE TRANSFER MITIGATION VIA TASK-AWARE ROUTING

Define the (per-token) gradient of expert $i$ on task $t$ as $g_{t,i}(x) = \nabla_{\theta_i} \mathcal{L}_t(E_i(x))$, and the *inter-task interference* at expert $i$ as

$$\Gamma_i = \mathbb{E}_x \Big[ \sum_{t \neq t'} \omega_i(x; e_t) \, \omega_i(x; e_{t'}) \, \langle g_{t,i}(x), g_{t',i}(x) \rangle \Big].$$

Large positive $\Gamma_i$ indicates conflicting supervision flowing through the same expert.

**Proposition 3** (Task-aware attenuation of interference). *Assume $G_{lang}(\bar{F}_v, e_t)$ is task-discriminative in the sense that $\mathbb{E}_x[\omega_i(x; e_t) \, \omega_i(x; e_{t'})] \leq \rho_i$ for $t \neq t'$ with $\rho_i \ll 1$. Then the total interference $\sum_i \Gamma_i$ is upper bounded by a constant proportional to $\sum_i \rho_i$. Moreover, if $G_{lang}$ becomes near one-hot over tasks (i.e., experts specialize), then $\rho_i \to 0$ and interference vanishes in the limit.*

*Sketch.* The dual gating multiplies token affinity $G_{tok}$ with a task prior $G_{lang}$. For $t \neq t'$, the product of routing weights at the same expert is suppressed by task-discriminativity, reducing the cross-terms $\langle g_{t,i}, g_{t',i} \rangle$ in expectation. Hence the bound scales with $\rho_i$, which shrinks as specialization emerges during training.

**Implication.** Language-guided global gating serves as a *task prior* that steers tokens towards experts aligned with the task semantics, thereby *reducing negative transfer* by lowering the probability that conflicting tasks co-route through the same expert.

**Proposition 4** (Consistency of language-guided routing). *Under the separability assumption and a cross-entropy objective on the router, gradient descent with sufficiently small step size yields routing probabilities $G_{lang,j}(\bar{F}_v, e_t) \to 1$ for $t \in T_j$ and $\to 0$ otherwise. Consequently, experts specialize to disjoint task groups in probability.*

*Sketch.* The router is a linear-softmax classifier over experts, driven by $(\bar{F}_v, e_t)$. With $e_t$ linearly separable by $\{\mu_j\}$, standard margin-based convergence analyses apply: the logits align with $\mu_j$, and softmax probabilities concentrate on the correct expert group. The residual dependence on $\bar{F}_v$ refines routing within a task via image-level context.

A.2.5   PUTTING IT TOGETHER

Props. 1–4 jointly explain the empirical phenomena observed with **LangSAM**: (i) the model family is expressive enough to realize task-conditional solutions while sharing experts when beneficial; (ii) sparse dual gating reduces the hypothesis complexity, improving generalization; (iii) language-guided routing injects a task prior that attenuates gradient interference and drives experts toward interpretable specialization. These effects align with our training dynamics: early iterations show near-uniform expert usage, while later stages exhibit peaked, task-specific routing distributions.

**language-guided MoE** Our method uses the CLIP text encoder to convert the task description into a vector (fixed dimension, such as 512). Then, this vector is used as the Task Routing Token and

input into the gating network to calculate the weight distribution of expert together with the visual features. $gating = (token\text{-}gate\text{-}weights[idx, top, None]) * (language\text{-}guided\text{-}weights[token[idx]])$, allowing the task text embedding to influence the selection of experts.

## A.3 EXPERIMENTAL DETAILS

### A.3.1 TEXT PROMPT

We construct task-specific text prompts based on detailed task descriptions and, when applicable, incorporate class label information from different datasets. The resulting task description embeddings serve as language-guided routing tokens in our framework. For clarity and reproducibility, we also provide the implementation of all task-specific text prompts used in our method.

Listing 1: Task prompts dictionary for dense prediction tasks.

```
self.task_prompts = {
    "semseg": "For each pixel, assign a semantic category label from a
        predefined set of object classes.",
    "depth": "For each pixel, estimate its continuous depth value
        representing the relative distance from the camera plane.",
    "human_parts": "For each pixel, assign a label corresponding to a
        specific human body part (e.g., head, torso, arms, legs).",
    "normals": "For each pixel, predict a normalized 3D orientation vector
        representing the local surface orientation in the scene.",
    "sal": "For each pixel, predict a saliency score indicating the pixel
        being part of a visually prominent region.",
    "edge": "For each pixel, detect and localize semantic boundaries that
        delineate distinct object regions in the image."}
```

A task description serves as the base text prompt. Using a vision–language model (*e.g.*, GPT-5), we further augment this prompt into richer, more informative variants.

Listing 2: Task prompts used for language-guided routing.

```
self.task_prompts = {
    "semseg": "Assign each pixel a semantic class label from the dataset
        vocabulary. Categories include indoor objects (chair, table, sofa,
         bed, cabinet, lamp, window, door), structural elements (wall,
        floor, ceiling), outdoor entities (road, sidewalk, sky, water,
        grass, tree, building), and movable objects (person, car, bus,
        bicycle, dog, cat).",
    "depth": "Predict for each pixel a continuous depth value relative to
        the camera plane, accurately modeling flat surfaces like floors,
        walls, and tables, while preserving discontinuities at object
        boundaries such as chairs, beds, and vehicles.",
    "human_parts": "Label each pixel with fine-grained part categories.
        For humans: head, torso, arms, legs, hands, feet, eyes, nose, and
        mouth. For vehicles and objects: parts such as wheels, doors,
        wings, engines, windows, and mirrors are included. For animals:
        torso, head, legs, tail, horns, wings, and beak.",
    "normals": "Estimate for each pixel a unit-length 3D surface normal
        vector in camera coordinates, capturing local surface orientation
        of indoor objects (tables, sofas), room structures (walls, floors)
        , and outdoor elements (roads, buildings, vegetation), while
        maintaining sharp discontinuities at edges and boundaries.",
    "sal": "Predict for each pixel a saliency score between 0 and 1 that
        highlights visually prominent foreground regions, including people
        , animals, vehicles, furniture, and objects of interest, while
        suppressing background clutter like sky, walls, or floors.",
    "edge": "Detect for each pixel whether it lies on a semantic boundary
        separating different objects or parts. Boundaries often occur
        between walls and furniture, between people and their surroundings
        , or between vehicles and road surfaces, and should be thin,
        precise, and topologically coherent."}
```

Table 6: Ablation study for text prompt variants.

| Text Prompt Variants | SemSeg (mIoU)↑ | Depth (rmse)↓ | Normal (mErr)↓ | Bound (odsF)↑ |
|---|---|---|---|---|
| Full method *w/* augment text prompt | 54.73 | 0.4646 | 16.68 | 79.20 |
| *w/o* augment text prompt | 54.36 | 0.4701 | 16.72 | 79.10 |

Table 6 shows that incorporating augmented text prompts consistently improves performance across all tasks. In semantic segmentation, the mIoU rises from 54.36 to 54.73, while depth estimation achieves lower error (0.4701 → 0.4646). Similarly, surface normal estimation and boundary detection both benefit slightly (16.72 → 16.68 and 79.10 → 79.20). Although the gains are modest, they demonstrate that prompt augmentation provides more informative task guidance, leading to more stable and effective multi-task representations.

## A.4 DISCUSSIONS

### A.4.1 MORE EXPERTS RESULTS

Table 7: Ablation study for expert numbers on NYUD-v2 dataset.

| Expert numbers | SemSeg (mIoU)↑ | Depth (rmse)↓ | Normal (mErr)↓ | Bound (odsF)↑ |
|---|---|---|---|---|
| 2 | 53.54 | 0.4889 | 16.92 | 78.9 |
| 4 | 53.60 | 0.4713 | 16.81 | 78.9 |
| 6 | 54.36 | 0.4701 | 16.72 | 79.1 |
| 8 | 53.28 | 0.4849 | 16.97 | 79.0 |
| 16 | 54.59 | 0.4783 | 16.88 | 79.2 |
| 32 | 54.61 | 0.4801 | 16.94 | 79.4 |

**Effect on expert numbers.** Table 7 evaluates the impact of varying expert numbers on NYUD-v2. Performance improves steadily from 2 to 6 experts, suggesting sufficient capacity is critical for task specialization. Beyond 6, gains saturate and fluctuate slightly, with 16 and 32 experts offering only marginal improvements. These results highlight that a moderate number of experts achieves the best balance between efficiency and effectiveness.

### A.4.2 TASK-LEVEL EXPERT DIVISION

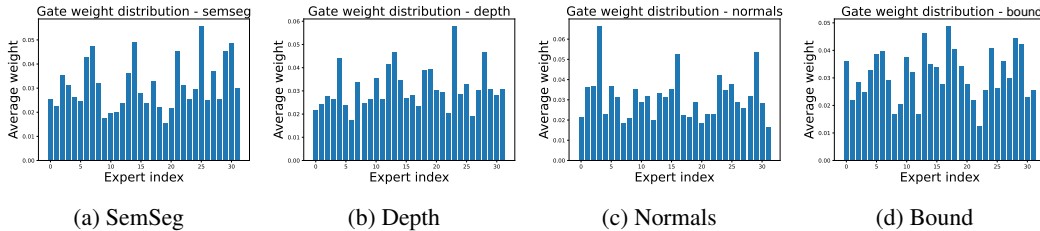

| (a) SemSeg | (b) Depth | (c) Normals | (d) Bound |
|---|---|---|---|

Figure 4: We set the number of experts to 32 (selected at 1 iteration). Expert routing distributions in **LangSAM** for SemSeg, Depth, Normals, and Bound. Different tasks activate distinct expert subsets, showing language-guided routing enables interpretable specialization and reduces task interference.

**Analysis of Gate Distribution Dynamics.** We further investigate how the gate distribution evolves during training. At the early stage (*e.g.*, iteration 1, Figure 4), the routing weights across different experts exhibit nearly uniform distributions, with no clear preference. This behavior is expected, as the router and experts are randomly initialized and the model has not yet learned to differentiate task-specific patterns. However, as training progresses (*e.g.*, iteration 2000, Figure 5), the distributions become increasingly sparse: only a small subset of experts receive consistently high weights (up to 0.8 on average), while others are largely suppressed. This transition highlights that

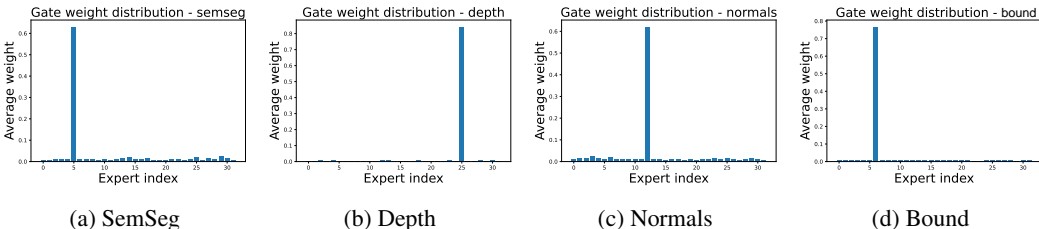

(a) SemSeg          (b) Depth          (c) Normals          (d) Bound

Figure 5: The distribution of experts after 2000 iterations. We set the number of experts to 32. Expert routing distributions in **LangSAM** for SemSeg, Depth, Normals, and Boundary detection.

the language-guided router gradually discovers meaningful task–expert correspondences, allowing different experts to specialize in distinct aspects of multi-task dense prediction. Such sparsification not only improves computational efficiency by avoiding redundant expert activation but also provides interpretability, as the emerging specialization indicates how LangSAM aligns experts with semantic or geometric task priors, thereby mitigating negative transfer across tasks.

### A.4.3    EARLY STOP STRATEGY

**Observation on Early Stop Strategy.** Table 8 reports the performance of LangSAM under different training iterations. We observe that the model exhibits rapid improvement in the early stage: at 2k iterations, the performance is poor across all tasks, while at 6k iterations, both segmentation (mIoU: 52.37) and boundary detection (odsF: 77.2) improve substantially. Performance continues to increase until around 16k iterations, where the model achieves the best overall trade-off across tasks (SemSeg: 54.38, Depth rmse: 0.4807, Normal mErr: 16.85, Bound: 79.2). After this point, additional training does not yield further gains and even causes minor fluctuations, suggesting the model risks overfitting or over-specializing experts. This indicates that an early stop around 16k iterations is sufficient for stable convergence, while prolonged training offers diminishing returns.

Table 8: Ablation study for 40k iterations.

| Iterations | SemSeg (mIoU)↑ | Depth (rmse)↓ | Normal (mErr)↓ | Bound (odsF)↑ |
|---|---|---|---|---|
| 2000 | 37.57 | 0.7040 | 18.23 | 52.5 |
| 6000 | 52.37 | 0.6756 | 17.27 | 77.2 |
| 10000 | 53.10 | 0.5337 | 16.98 | 78.1 |
| 16000 | 54.38 | 0.4807 | 16.85 | 79.2 |
| 20000 | 53.88 | 0.4829 | 16.86 | 79.1 |
| 26000 | 53.24 | 0.4737 | 16.91 | 78.9 |
| 30000 | 52.99 | 0.4674 | 16.98 | 79.0 |
| 36000 | 53.31 | 0.4693 | 16.95 | 79.1 |
| 40000 | 53.16 | 0.4665 | 16.94 | 79.3 |

### A.5    VISUALIZATIONS

### A.5.1    GATING HEATMAP

**Analysis of global gating behavior.** As shown in Figure 6, to better understand the behavior of the proposed LangSAM framework, we analyze the gating heatmaps across four representative dense prediction tasks: semantic segmentation, depth estimation, bound detection, and surface normal prediction. Each heatmap reports the averaged gating weights over eight experts, thus revealing which experts are preferentially activated by each task.

**Semantic Segmentation.** Semantic segmentation relies heavily on Expert 0 (0.25), substantially larger than any other expert, with only modest support from Expert 4 (0.14). This bias suggests the emergence of a semantic-specialized expert that aggregates contextual information across categories, while other experts remain underutilized.

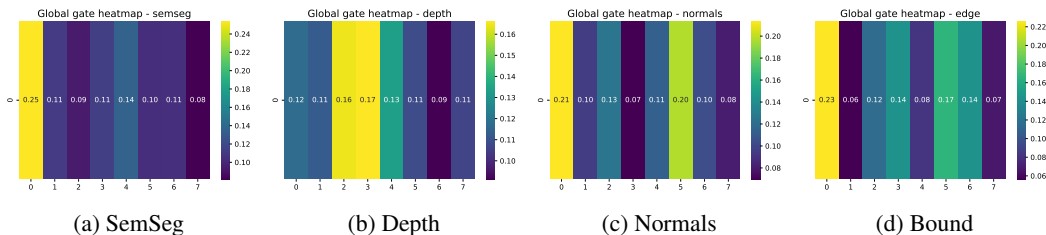

(a) SemSeg       (b) Depth       (c) Normals       (d) Bound

Figure 6: We set the number of experts to 8. Expert gating heatmap in **LangSAM** for 4 tasks.

**Depth Estimation.** The routing distribution is moderately spread, with Expert 2 (0.16) and Expert 3 (0.17) receiving the highest weights, followed by Experts 0/1/4/5 (0.11-0.13). This suggests that depth estimation benefits from a combination of experts capturing multi-scale geometric cues rather than relying on a single dominant expert.

**Surface Normal.** Normal also demonstrates a clear geometric preference: Expert 0 (0.21) and Expert 5 (0.20) dominate, whereas others play secondary roles (0.07–0.13). This strongly parallels the depth case, confirming that Experts 0 and 5 encode structural priors useful for modeling geometry.

**Bound Detection.** In contrast, edge detection exhibits a more skewed distribution, with Expert 0 (0.23) and Expert 5 (0.17) contributing most strongly, while Experts 1 and 7 remain minimally activated. This indicates that a small subset of experts specialize in local contrast and contour-sensitive representations, which are crucial for boundary localization.

Two key insights can be observed. First, Expert 0 consistently receives high weights across all tasks, acting as a *generic expert* that captures universally useful features. Second, Expert 5 is consistently activated in geometry-related tasks (depth and normals), while edge detection additionally engages Expert 6. Semantic segmentation, in contrast, strongly concentrates on a single semantic expert (Expert 0). These patterns validate the desired *task-specific specialization with shared expert reuse*: tasks dynamically recruit distinct subsets of experts while still leveraging common ones, striking a balance between specialization and generalization.

### A.5.2 VISUALIZATION COMPARISON

As shown in Figures 7 and 8, the qualitative comparisons on PASCAL-Context (5 tasks) and NYUD-v2 (4 tasks) clearly demonstrate the advantages of our framework in multi-task dense prediction. On NYUD-v2, our model produces sharper semantic boundaries, more consistent depth maps, and accurate surface normal orientations, highlighting its ability to capture both global structures and fine-grained geometric details. On PASCAL-Context, the visualizations reveal that our method not only preserves category-level semantics across diverse object classes but also maintains coherent boundary delineation and part consistency, even in cluttered scenes.

These visualizations highlight the strength of language-guided MoE in balancing shared representations with task-aware specialization, enabling both semantic fidelity and geometric consistency across dense prediction tasks.

### A.6 EXPLORATION WITH LARGE MODELS

**Cross-modal and temporal generalization.** Large video and vision–language models open MTL beyond still images. Extending LangSAM to spatiotemporal backbones (*e.g.*, SAM 2-like video encoders) with *time-aware routing* may enable joint learning of optical flow, depth, tracking, and segmentation. Language priors can disambiguate long-range correspondences and specify temporal tasks (*i.e.*, track the person and refine boundaries over time).

**Future work.** LangSAM demonstrates that language-guided expert routing can inject task-aware into powerful but task-agnostic backbones (*e.g.*, SAM 2). We anticipate a convergence of instruction-tuned vision–language backbones, scalable sparse MoE, and compositional routing, yielding MTL systems that are *open-vocabulary*, *compute-efficient*, and *interpretable*—capable of following natural-language goals to perform diverse dense predictions with minimal retraining.

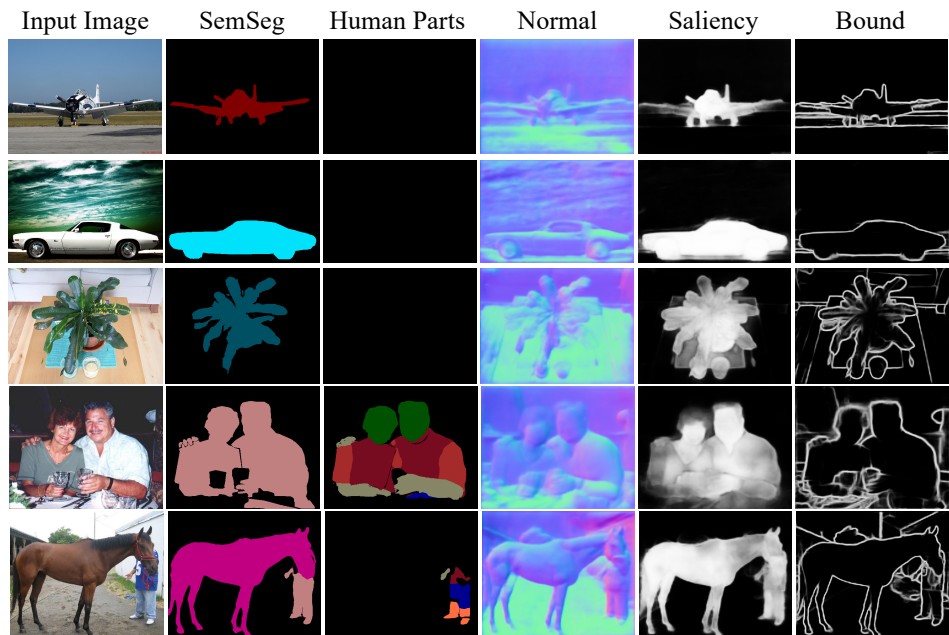

Figure 7: Qualitative results on 5-task PASCAL-Context dataset. The results illustrate the effectiveness of our model in capturing both semantic and fine-grained details. Best viewed in color and zoom.

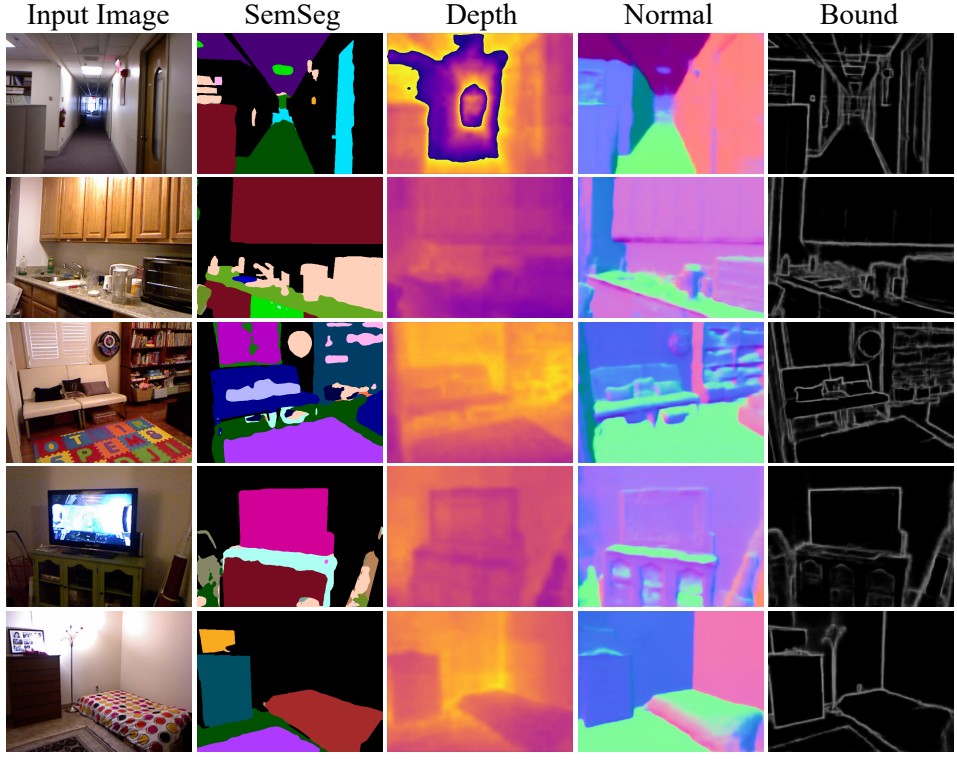

Figure 8: Qualitative results on 4-task NYUD-v2 dataset. The results illustrate the effectiveness of our model in capturing both semantic and fine-grained details. Best viewed in color and zoom.

