# OpenReview forum: "LangSAM: Language-Guided Expert Routing on SAM2 for Dense Scene Understanding"
_ICLR.cc/2026/Conference — ICLR 2026 Conference Withdrawn Submission_

### Official Review · Reviewer_91KG · 2025-10-29

**Soundness:** 1
**Presentation:** 2
**Contribution:** 2
**Rating:** 2
**Confidence:** 4

**Summary:**

This paper proposes LangSAM that leverages a language-guided Mixture-of-Experts (MoE) architecture built on top of a frozen SAM 2 backbone. It features a language-guided routing mechanism. Instead of relying solely on visual features to select experts, LangSAM also uses natural language task descriptions (e.g., "predict a normalized 3D orientation vector..."). These prompts are encoded by a frozen CLIP model to produce task embeddings. LangSAM features a dual-gated mechanism: A standard token-level visual router selects experts based on local image features. A language-guided router fuses the global visual features with the task embedding to generate task-level weights. The final expert activation is a product of both gates, allowing experts to specialize both spatially and semantically by task. The architecture consists of task-specific MoE blocks for coarse predictions and a shared MoE block for cross-task refinement.

The authors evaluate LangSAM on the NYUD-v2 and PASCAL-Context datasets across six dense prediction tasks. Results show that LangSAM consistently outperforms strong baselines and recent multi-task learning (MTL) methods, achieving a superior overall multi-task performance trade-off ($\Delta_m$) while being parameter-efficient. Ablation studies and visualizations confirm that the language-guided routing leads to interpretable expert specialization.

**Strengths:**

- The paper conducts extensive experiments covering many dense prediction tasks on two popular multi-task learning benchmarks.

- The writing is clear and easy to understand.

**Weaknesses:**

- I have a fundamental concern about the proposed method. How can language embeddings assist the router network in determining which expert activations to use? The language embeddings come from a completely different domain than the router network’s inputs and outputs, so it’s unclear to me why they would be theoretically helpful.

- Why was the SAM backbone chosen? SAM does not perform well on semantic understanding tasks such as semantic segmentation, as also evidenced in Table 1.

**Questions:**

- Please answer my questions in the weakness part.

---

### Official Review · Reviewer_zAQJ · 2025-10-30

**Soundness:** 2
**Presentation:** 2
**Contribution:** 2
**Rating:** 4
**Confidence:** 3

**Summary:**

This paper proposes LangSAM, a language guided MoE framework for scene understanding based on SAM2. It uses language prompts to guide the activation of experts on different tasks that allows fine-grained task-aware feature representation. The proposed dual MoE architecture has a task-specific language router and shared MoE, which balances specialization and generalization.

Experiments were conducted on NYUD-v2 and PASCAL-Context to validate the effectiveness of the proposed method.

**Strengths:**

1. The motivation of this paper is strong. Multi-task dense scene understanding is a highly relevant and challenging area: combining segmentation, depth, normals, etc on the same model is nontrivial and worth studying.
2. Experiments are comprehensive - The paper conducts experiments on multiple datasets and demonstrate the results on various tasks (segmentation, part segmentation, normals, etc).
3. Presentation is decent - the figures are well designed that help readers understand the architecture.

**Weaknesses:**

1. The main claimed contribution is the use of language to route visual expert via soft conditioning. But it only affect dense per-pixel prediction indirectly and we don't see a significant performance leap.
2. The paper claims that unlike previous methods that work on fixed set of tasks, this work is flexible and scalable. But all benchmarks still evaluate fixed tasks (“depth”, “normals”, “semseg”) — they don’t test language-driven new task generalization.

**Questions:**

1. It doesn't seem like adding language conditioned router affect the results much. Can you highlight the biggest contribution and the most significant experimental result?
2 Besides performance, are there any potential of this work? Can it enable a new type of segmentation task beyond the fixed-set via language description?

---

### Official Review · Reviewer_SB4C · 2025-10-31

**Soundness:** 2
**Presentation:** 3
**Contribution:** 2
**Rating:** 4
**Confidence:** 4

**Summary:**

The paper introduces LangSAM, a language-guided mixture-of-experts (MoE) architecture built on a frozen SAM2 backbone for multi-task dense prediction. It combines token-level visual gating with a language-guided global router, using CLIP text embeddings of task descriptions to route features through expert networks. LangSAM includes both task-specific and shared MoE blocks, achieving state-of-the-art results on NYUD-v2 and PASCAL-Context across segmentation, depth, saliency, and boundary tasks.

**Strengths:**

- Good empirical gains on standard dense-prediction MTL benchmarks. LangSAM shows consistent improvements (reported ∆m gains and per-task metrics) on both PASCAL-Context and NYUD-v2, and competitive fine-tuning numbers under a frozen SAM2 backbone compared to modern baselines (TaskExpert, TaskPrompter, MLoRE, MTSAM). The tables show improved tradeoffs and in many cases better or comparable per-task metrics with lower FLOPs/params.
- Detailed ablations and interpretability analyses. The paper includes ablation studies that isolate task-specific vs shared MoE components, presence/absence of the language-guided router, and the effect of expert count; it also visualizes gating distributions over training showing emergent specialization, supporting their claims about reduced interference / interpretable experts.

**Weaknesses:**

- Limited novelty. While using language as a prior is appealing, related ideas exist [1, 2]. Further, the router design of LangSAM is highly similar to that of TaskExpert [3].
- Efficiency / overhead tradeoffs need clearer presentation. Tables report FLOPs/params, but the exact runtime / memory behavior during training and inference (with different top-k settings, expert counts, and the addition of the language router) is not fully quantified. Sparse MoE can introduce communication/implementation costs; more practical profiling (latency, GPU memory, throughput) would help assess deployability.
- Theoretical claims need empirical linkage. Appendix contains propositions about expressivity, Rademacher complexity, and interference attenuation. While plausible, the experimental section does not fully validate these theoretical predictions quantitatively (e.g., measured gradient conflicts, a formal study linking gate sparsity to generalization). Strengthening the connection between theory and empirical observations would make the argument more convincing.
- Paper writing: High-dimensional tensor symbols should be written in bold, while scalar symbols should not be bolded. However, the use of symbols in the paper is inconsistent.

[1] Multi-Task Dense Prediction via Mixture of Low-Rank Experts. CVPR 2024.

[2] ClipSAM: CLIP and SAM collaboration for zero-shot anomaly segmentation. Neurocomputing 2025.

[3] Taskexpert: Dynamically assembling multi-task representations with memorial mixture-of-experts. ICCV 2023.

**Questions:**

- Baselines & protocol parity: For all baselines in Tables 1–3 (TaskExpert, TaskPrompter, MLoRE, MTSAM, etc.), were they re-implemented under the same frozen SAM2 backbone and identical optimization/hyperparameter budget, or are those numbers taken from the original papers with different backbones? Please clarify and (if not already done) provide matched-backbone re-runs or clearly label mixed-protocol numbers.
- Prompt sensitivity: How sensitive are results to the exact wording of the CLIP prompts? Please provide an ablation where you (a) use very short terse prompts, (b) use the more verbose augmented prompts, and (c) perturb wording (synonym swaps). Report per-task metric variance to show robustness.
- Gradient interference measurement: The theory claims reduced gradient interference via language priors. Do you have direct measurements that substantiate the claimed reduction? If so, please add a short quantitative experiment.

---

### Official Review · Reviewer_VjxW · 2025-11-01

**Soundness:** 2
**Presentation:** 2
**Contribution:** 2
**Rating:** 2
**Confidence:** 3

**Summary:**

The paper proposes LangSAM, a language-guided mixture-of-experts (MoE) framework built on top of the SAM2 foundation model for multi-task dense scene understanding. The core idea is to use natural language task descriptions encoded via a frozen CLIP text encoder as explicit routing signals to condition expert activation in an MoE architecture. LangSAM introduces a dual-gated routing mechanism that combines token-level visual gating with global language-guided gating, enabling both spatial and semantic specialization. The framework includes both task-specific MoE blocks for coarse predictions and a shared MoE block to model cross-task dependencies. Evaluated on NYUD-v2 and PASCAL-Context across six dense prediction tasks (e.g., semantic segmentation, depth, normals), LangSAM consistently outperforms strong SAM2-based baselines and recent multi-task learning methods, while offering interpretable expert routing behaviors that align with task semantics (e.g., geometry-focused vs. semantics-focused experts).

**Strengths:**

- The paper successfully leverages frozen CLIP-encoded task descriptions to guide MoE expert selection, enabling semantically meaningful and interpretable task-specific specialization (e.g., geometry vs. semantics).
- The paper shows consistent quantitative gains over several strong baselines on two benchmarks across six dense prediction tasks.

**Weaknesses:**

1. LangSAM is a well-engineered but incremental contribution that combines existing ideas (MoE + CLIP prompts + SAM2) without deep innovation. The core idea of using CLIP-encoded natural language prompts to condition MoE routing is not fundamentally novel. Previous paradigms like TaskPrompter and TaskExpert already integrate language/task prompts into multi-task or MoE architectures. LangSAM essentially replaces hand-coded task IDs with frozen CLIP embeddings—a useful engineering choice, but not a new paradigm. Moreover, the language signal is used only as a static task embedding, not for compositional reasoning, open-vocabulary generalization, or instruction following. This reduces “language guidance” to a fancy initialization of task vectors, undermining the paper’s central narrative.
2. The experimental gains are modest and potentially inflated by unfair baselines. Cross-backbone comparisons lack fairness: In Table 1, LangSAM (SAM2-H-L) is compared against methods using HRNet18 or ViT-L. Since SAM2 is a stronger backbone, performance gains may stem from the encoder, not the proposed routing.
3. No ablation on CLIP vs. learnable prompts: The paper never tests whether learnable task embeddings (without CLIP) perform similarly, which would undermine the necessity of language.
4. The method heavily relies on carefully curated prompts, but provides no analysis of robustness:
- What if prompts are noisy, ambiguous, or generated by an LLM?
- Can the model generalize to unseen tasks (e.g., “predict reflectance”) via zero-shot routing?
- How sensitive is performance to prompt phrasing? (e.g., “estimate depth” vs. “infer how far each pixel is”)
Without such analysis, the claim of “language as a universal interface” is unsubstantiated. The method’s reliance on handcrafted prompts and frozen backbones limits its practicality and generalizability.
5. The dual-gating mechanism (token-level + language-guided) is not rigorously justified. Figure 3 and Figure 6 show that routing is dominated by the language gate (e.g., SemSeg always activates expert 0), suggesting the token-level gate is redundant.
6. The theoretical analysis (Appendix A.2) consists of unproven propositions with only sketch-level arguments. For example, Proposition 3 claims language routing reduces interference, but no quantitative measure of Γi (interference) is provided.
7. While the paper claims “fewer FLOPs and parameters” (Table 1), it ignores real-world efficiency:
- MoE sparsity is illusory in dense prediction: Every pixel activates experts, so even with top-k=2, computation is nearly dense.
- No latency, memory, or throughput measurements are reported—critical for MoE systems.
- CLIP text encoding overhead is ignored (though cached, it adds latency in dynamic task settings).
 Thus, the “efficient sparse architecture” claim is not empirically validated.

**Questions:**

My major concerns are outlined in the "Weaknesses" part.

---

### Note · Authors · 2025-11-16

I have read and agree with the venue's withdrawal policy on behalf of myself and my co-authors.